# Association of Plant-Based Diet Indices and Abdominal Obesity with Mental Disorders among Older Chinese Adults

**DOI:** 10.3390/nu15122721

**Published:** 2023-06-12

**Authors:** Ran Qi, Baihe Sheng, Lihui Zhou, Yanchun Chen, Li Sun, Xinyu Zhang

**Affiliations:** 1School of Public Health, Tianjin Medical University, Tianjin 300070, China; qiran@tmu.edu.cn (R.Q.); shengbaihe@tmu.edu.cn (B.S.); zlhkindle@tmu.edu.cn (L.Z.); chenyanchun@tmu.edu.cn (Y.C.); 2School of Nursing, Tianjin Medical University, Tianjin 300070, China; sunli_12402@tmu.edu.cn

**Keywords:** Chinese Longitudinal Healthy Longevity Survey, plant-based diet indices, abdominal obesity, depression, anxiety

## Abstract

We aimed to explore the correlation between plant-based diet indices and abdominal obesity with depression and anxiety among older Chinese adults. This study used a cross-sectional design using data from the 2018 Chinese Longitudinal Healthy Longevity Survey (CLHLS). We used a simplified food frequency questionnaire to evaluate the overall plant-based diet index (PDI), the healthful plant-based diet index (hPDI), and the unhealthful plant-based diet index (uPDI) separately, based on the potential health effects of the foods. Waist circumference (WC) was used to define abdominal obesity. The 10-item Center for Epidemiologic Studies Depression Scale (CES-D-10) and the 7-item Generalized Anxiety Disorder Scale (GAD-7) were applied to estimate depression symptoms and anxiety symptoms, respectively. Multi-adjusted binary logistic regression models were conducted to explore the effects of the three plant-based diet indices, abdominal obesity status, and their interaction on depression and anxiety. We enrolled a total of 11,623 participants aged 83.21 ± 10.98 years, of which 3140 (27.0%) participants had depression and 1361 (11.7%) had anxiety. The trend in the prevalence of depression/anxiety across increasing quartiles of the plant-based diet indices was statistically significant after controlling for potential confounders (*p*-trend < 0.05). Abdominal obesity was related to a lower prevalence of depression (OR = 0.86, 95% CI: 0.77−0.95) and anxiety (OR = 0.79, 95% CI: 0.69−0.90) compared with non-abdominal obesity. The protective effects of the PDI and hPDI against depression (OR = 0.52, 95% CI: 0.41−0.64; OR = 0.59, 95% CI: 0.48−0.73, respectively) and anxiety (OR = 0.75, 95% CI: 0.57−1.00; OR = 0.52, 95% CI: 0.39−0.70, respectively) were more pronounced in non-abdominally obese participants. The harmful effects of the uPDI against depression (OR = 1.78, 95% CI: 1.42−2.23) and anxiety (OR = 1.56, 95% CI: 1.16−2.10) were more pronounced in non-abdominally obese participants. In addition, a significant interaction between the plant-based diet indices and abdominal obesity was observed in terms of causing the prevalence of depression and anxiety. Consuming more of a healthful plant-based diet and less of an animal-based diet is related to a lower prevalence of depression and anxiety. A healthful plant-based diet plays a vital role in non-abdominally obese individuals.

## 1. Introduction

The trend of global population aging is increasing. It is estimated that by 2030 there will be more than 1 billion elderly people [1], and aging is often accompanied by a series of physical and mental problems [2]. Depression is a widespread mental illness, afflicting about 258 million people worldwide [3]. It is also one of the major diseases in the elderly population, with the prevalence of depression showing a growing trend [2]. Depression is one of the main reasons for emotional distress in later life, and it also significantly affects the health of older people and reduces their quality of life [4]. Anxiety, a very common psychiatric disorder in the context of depression, has a global prevalence of 7.3% [5]. Nearly two-thirds of patients with major depression experience clinical anxiety [6], which greatly increases people’s demand for medical care as well as individual and social costs [7]. Existing data indicate that mental health disorders have become a serious threat to public health [8,9]. Therefore, proactive interventions to prevent depression and anxiety are essential to improve the health status of elderly populations and reduce the financial burden on individuals and society.

Diet is an important modifiable behavioral factor, and a healthful diet can take a positive role in the development and intervention of depression and anxiety [10,11,12]. A study showed that people without depression tend to ingest more beans, vegetables, fruits, and fewer desserts than those with depression [13]. As one of the world’s most recognized healthy diet patterns, the Mediterranean diet has been shown to ameliorate symptoms manifested by patients with depression [14,15]. Moreover, the Western dietary pattern characterized by a high intake of calories, fat, and protein has been shown to be adversely associated with depression and anxiety [16]. However, focusing attention only on individual foods is one-sided. Dietary patterns are often closely related to the cultural characteristics and eating habits of the population, which is difficult to be completely duplicated in other populations [11]. Chinese dietary patterns have undergone considerable changes over the past few decades, but there are also differences between regions [17,18]. Plant-based diet indices avoid these restrictions mentioned above through an integrated way of taking into account the intake of plant-based and animal-based foods [19,20,21]. Plant-based diet indices include the overall plant-based diet index (PDI), the healthful plant-based diet index (hPDI), and the unhealthful plant-based diet index (uPDI), each of which has a slightly different focus on food intake. Previous studies have demonstrated that plant-based diet indices are related to a series of adverse outcomes, including cognitive impairment, chronic disease, and all-cause mortality [19,22,23,24]. However, research discussing the correlations between plant-based diet indices and depression and anxiety in elderly populations is still lacking.

Obesity, especially abdominal obesity, has been proven to adversely affect physiological function and increase morbidity and mortality [25,26,27]. However, previous epidemiological research has demonstrated contradictory results regarding the relationship between obesity and depression and anxiety. Some research suggests that obesity contributes to poor mental health [28,29,30], and other studies have observed that abdominal obesity can create a positive effect on depression and anxiety [31,32,33,34,35]. Compared with other obesity indicators, waist circumference (WC) has been proven to be a representative indicator of abdominal visceral adipose tissue deposition due to its focus on measuring body fat [36]. Therefore, it is of practical importance to use WC in measuring abdominal obesity. To date, the specific associations between abdominal obesity and depression and anxiety warrant further investigation. Few studies have reported the potential interaction between plant-based diet indices on the basis of different abdominal obesity statuses and mental disorders.

In this paper, we aimed to assess the correlation between plant-based diet indices and abdominal obesity with depression and anxiety using representative data of older Chinese adults. Furthermore, we investigated the potential association between abdominal obesity and three plant-based diet indices with respect to depression and anxiety through stratified analyses.

## 2. Methods

### 2.1. Data Sources and Study Population

In order to explore the association of plant-based diet indices and abdominal obesity with depression and anxiety, we used cross-sectional data from the 2018 Chinese Longitudinal Healthy Lifespan Survey (CLHLS) for examination. The CLHLS, which collects data utilizing a questionnaire survey, was established in 1998, and it targets elderly people aged 65 years and above. The CLHLS uses a multi-stage stratified sampling method and covers 23 provinces in China, accounting for approximately 85% of the Chinese population. Meanwhile, face-to-face questionnaires were administered to participants to ensure a good response rate. More detailed descriptions of the CLHLS design can be found elsewhere [37,38,39]. The CLHLS was approved by the Biomedical Ethics Committee of Peking University, China (IRB00001052–13074), and all participants signed informed consent.

A total of 15,874 participants completed the CLHLS in 2018. An amount of 84 participants were excluded because they were younger than 65 years of age. We further excluded 793 participants with missing information on diet, 3173 participants with missing data on depression and anxiety, and 162 participants with missing data on WC. We also excluded 39 participants with extreme measurement values (WC < 45 cm, WC > 150 cm) [40]. A total of 11,623 participants were included in the analyses. Figure 1 shows a detailed flowchart of the selection of participants for this study.

### 2.2. Assessment of Abdominal Obesity

WC was used to determine abdominal obesity. When measuring waist circumference, participants were asked not to inhale, and the interviewer would record the waist circumference by applying a tape measure directly against the skin. According to the criteria used in existing studies [31], abdominal obesity was defined by WC ≥ 80 cm in women and WC ≥ 85 cm in men.

### 2.3. Calculation of Plant-Based Diet Indices

Participants reported the types and frequency of foods they consumed in a simplified food frequency questionnaire. A total of 16 food groups were included in this study for assessment. We classified these food groups into three types based on the potential health effects of the foods, namely, plant-based foods (including healthful plant-based foods and unhealthful plant-based foods) and animal-based foods. Whole grains, vegetable oils, fruits, vegetables, legumes, garlic, nuts, and tea were classified as healthful plant-based foods. Refined grains, preserved vegetables, and sugar were classified as unhealthful plant-based foods. Animal fats, meat, fish and aquatic products, eggs, and milk and dairy products were classified as animal-based foods.

Based on previous studies [22,23,41], we calculated the PDI, hPDI, and uPDI based on participants’ self-reported dietary consumption, with each food assigned a score range of 1 to 5. The PDI highlights the status of the overall diet, which focuses on dietary adherence to high plant-based food intake and low animal-based food intake. The hPDI emphasizes dietary compliance with a high intake of healthful plant-based foods. On the contrary, the uPDI is concerned with dietary adherence to a high intake of unhealthful plant-based foods [42]. For the PDI, the higher the intake of healthful and unhealthful plant-based foods, the higher the score (5 points for the most frequent consumption, 1 point for rarely or never). For the hPDI, the higher the intake of healthful plant-based foods, the higher the score; unhealthful plant-based foods were scored inversely (the higher the intake, the lower the score). For the uPDI, a higher frequency of consuming unhealthful plant-based foods was assigned a higher score; healthful plant-based foods were scored inversely. The scoring rule for animal-based foods was consistent, such that the higher the frequency of consumption, the lower the score. More information on the food groupings and scores can be found in Appendix A. In this study, the total score for each plant-based diet index ranged from 16 to 80. We further grouped the participants into four groups (Q1, Q2, Q3, Q4) according to the quartile level of their scores.

### 2.4. Assessment of Depression and Anxiety

The 10-item Center for Epidemiologic Studies Depression Scale (CES-D-10) was employed to measure depressive symptoms, which consists of 10 questions, each of which is divided into five levels of scoring. For negative state questions and questions asking about sleep quality, responses were scored as follows: “rarely or never” as 0, “sometimes” as 1, “often” as 2, and “always” as 3. Positive status questions were scored in reverse. The scale has a theoretical score range of 0–30, and in this study, a score ≥ 10 indicated depression.

The 7-item Generalized Anxiety Disorder Scale (GAD-7) was employed to measure anxiety symptoms, which consists of seven questions, each with four scoring levels. Responses of “never,” “occasionally,” “over half the time,” and “almost daily” were scored 0, 1, 2, and 3, respectively. The scale has a theoretical score range of 0 to 21, and a score ≥ 5 indicates anxiety. Previously, the questionnaires for the Chinese language used in CLHLS have been widely accepted and used [31,43,44]. In this paper, the Cronbach-α coefficients for the CES-D-10 and the GAD-7 were 0.805 and 0.918, respectively, and the KMO coefficients were 0.871 and 0.927, respectively, indicating good reliability and validity of the scales.

### 2.5. Assessment of Covariates

Demographic characteristics, socioeconomic characteristics, health behaviors, and physical status of the study population were assessed as covariates in this study. Demographic characteristics that were adjusted in this study included age (years), sex (male or female), and type of residence (urban or rural). Socioeconomic characteristics included marital status (married and cohabiting or other), cohabitation status (solitude or not living alone), education (illiterate, primary or secondary and above), occupation (agricultural or others), and economic situation (wealthy or not wealthy). Health behaviors included sleep duration (≤6 h, 7–8 h, or ≥9 h), smoking status (current, former, or never), alcohol consumption (current, former, or never), physical exercise (yes or no), frequent participation in sedentary leisure activities (yes or no), and frequent participation in active leisure activities (yes or no). Physical status included chronic disease (yes or no), comorbidity (yes or no), and body mass index (BMI; underweight, normal, overweight, or obese).

Participation in leisure activities was evaluated using eight activities: Tai Chi, square dancing, gardening, reading books/newspapers, raising domestic animals/pets, playing cards/mahjong, watching TV or listening to the radio, and participating in organized social activities. Among these, reading books/newspapers, playing cards/mahjong, and watching TV or listening to the radio were regarded as sedentary leisure activities; the other five were regarded as active leisure activities. The responses for participation in leisure activities were divided into five options, with responses of “almost every day” considered frequent participation and the remaining responses considered infrequent participation in leisure activity. Whether a participant engages in leisure activities frequently is defined by participation in any one or more leisure activities. Chronic disease prevalence was assessed by considering six common physical diseases in the elderly population (hypertension, diabetes, heart disease, stroke and cerebrovascular disease, respiratory disease, and cancer), which were determined by self-report. Comorbidity was defined as having two or more chronic diseases at the same time.

### 2.6. Statistical Analysis

Descriptive statistics were used to compare differences in the prevalence of depression and anxiety in participants with different characteristics. Multivariate binary logistic regression models were constructed to assess the associations of three plant-based diet indices and abdominal obesity with depression and anxiety, where the lowest quartile group of each plant-based diet index and the non-abdominal obesity group were selected as reference groups, respectively. For the purpose of controlling the effect of potential confounders on the study results, we used several models with different covariates for adjustment in this study. Model 1 controlled for participants’ demographic characteristics (age, sex, and residence). Model 2 further adjusted for socioeconomic characteristics (marital status, cohabitation status, education, occupation, and economic situation). Model 3 further controlled for health behaviors and physical status (sleep duration, smoking status, alcohol consumption, physical exercise, leisure activity participation, BMI, and the presence of chronic diseases and comorbidities). Multiple imputations by chained equations were conducted to fill in missing values of the covariates. We stratified the participants according to abdominal obesity status to assess whether the effects of PDI, hPDI, and uPDI on depression and anxiety were influenced by abdominal obesity status. The interactions were calculated by adding interaction terms for the PDI, hPDI, and uPDI with abdominal obesity in the fully adjusted logistic regression models. Fully adjusted models were also used for subgroup analyses to investigate whether the association between abdominal obesity and plant-based diet indices and two mental disorders differed by sex, age (≤80 years, >80 years), residence, and economic status.

IBM SPSS version 20.0 and R version 4.2.1 were used to conduct the statistical analysis of this study. GraphPad version 8.3 was used to visualize the results of the stratified analyses. The two-sided test *p* < 0.05 was considered statistically significant.

## 3. Results

A total of 11,623 participants were enrolled in this study with an average age of 83.21 ± 10.98 years, and 6197 (53.3%) participants were female. Overall, the results showed that 3140 (27%) and 1361 (11.7%) participants had depression and anxiety, respectively (Table 1). Moreover, 983 (8.5%) participants had both depression and anxiety. Mean scores on the PDI, hPDI, and uPDI among participants were 48.11 ± 5.50, 46.66 ± 5.37, and 49.13 ± 6.92, respectively. Participants with abdominal obesity accounted for 62.5% of the overall study population. Participants with depression were older compared with participants who did not have depression. Participants with depression and anxiety were more likely to be women, living in a rural area, not living with their spouse or without a spouse, living alone, having a low level of education, working in agriculture, not wealthy, having a short sleep duration, and those who never smoke, never drink alcohol, and never exercise, as compared with participants who did not have depression or anxiety. Participants with depression and anxiety also tended to report a higher prevalence of chronic diseases and comorbidities and were less likely to participate in leisure activities.

Table 2 and Table 3 show the multivariable-adjusted odds ratios (ORs) and 95% confidence intervals (CIs) of depression and anxiety in quartiles of the PDI, hPDI, and uPDI, respectively. The trend in the incidence of depression/anxiety across increasing quartiles of the plant-based diet indices was statistically significant in all models (*p*-trend < 0.05). In the fully adjusted model, participants in the highest quartile of the PDI and hPDI were more likely to have a lower prevalence of depression compared with those in the lowest PDI and hPDI quartile (OR = 0.61, 95% CI: 0.54–0.70; OR = 0.62, 95% CI: 0.54–0.71, respectively). Participants in the highest quartile of the uPDI showed a significantly higher prevalence of developing depression symptoms compared with those in the lowest uPDI quartile (OR 1.55, 95% CI: 1.35−1.78). However, the effect of the plant-based diet indices on depression and anxiety was attenuated in the fully adjusted model compared to the crude model.

In the fully adjusted model, participants in the highest quartile of the PDI and hPDI were more likely to have a lower prevalence of anxiety compared with those in the lowest PDI and hPDI quartile (OR = 0.81, 95% CI: 0.68−0.96; OR = 0.66, 95% CI: 0.56−0.79, respectively). A higher uPDI was consistently associated with a higher prevalence of developing anxiety compared with the lowest uPDI quartile, and participants in the highest quartile of the uPDI had the highest relative prevalence of developing anxiety (OR = 1.50, 95% CI: 1.25−1.80) (Table 3).

Table 4 demonstrates the association between abdominal obesity and depression or anxiety. The results showed that in the crude model, participants with abdominal obesity had a 21% (95% CI 0.72−0.86, *p* < 0.001) lower prevalence of depression and a 25% (95% CI 0.67−0.84, *p* < 0.001) lower prevalence of anxiety compared with those who did not have abdominal obesity. After further controlling for all covariates, abdominal obesity was still associated with a lower prevalence of depression (OR = 0.79, 95% CI: 0.72–0.87, *p* < 0.001) and anxiety (OR = 0.75, 95% CI: 0.66–0.85, *p* < 0.001) when compared with participants who did not have abdominal obesity.

After stratifying the population according to abdominal obesity status, we investigated the association of quartile membership of PDI, hPDI, and uPDI with depression and anxiety (Figure 2). Specifically, the odds ratio for depression was lower by 33% (OR = 0.68, 95% CI: 0.58–0.81, *p* < 0.001) in the highest quartile of the PDI in participants with abdominal obesity compared to the lowest quartile and by 49% (OR = 0.51, 95% CI: 0.41−0.64, *p* < 0.001) in the highest quartile of the PDI in participants without abdominal obesity compared to the lowest quartile. Among participants with abdominal obesity, the odds ratio for depression was lower by 36% (OR = 0.64, 95% CI: 0.54−0.75, *p* < 0.001) in the highest quartile of the hPDI compared to the lowest quartile and by 41% (OR = 0.59, 95% CI: 0.48−0.73, *p* < 0.001) in those without abdominal obesity. Among participants with abdominal obesity, the odds ratio for depression was higher by 43% (OR = 1.43, 95% CI: 1.20−1.71, *p* < 0.001) in the highest quartile of the uPDI compared to the lowest quartile and by 80% (OR = 1.80, 95% CI: 1.44−2.25, *p* < 0.001) in those without abdominal obesity. The association between PDI scores and anxiety was not significant in those with abdominal obesity (*p* > 0.05); among participants without abdominal obesity, those in the top quartile of the PDI had a 25% (OR = 0.75, 95% CI: 0.56−0.99, *p* < 0.05) lower odds ratio of anxiety compared to the lowest quartile. Among participants with abdominal obesity, the odds ratio for anxiety was lower by 23% (OR = 0.77, 95% CI: 0.61−0.96, *p* < 0.05) in the highest quartile of the hPDI compared to the lowest quartile and by 48% (OR = 0.52, 95% CI: 0.39−0.70, *p* < 0.001) in those without abdominal obesity. Among participants with abdominal obesity, the odds ratio for anxiety was higher by 46% (OR = 1.46, 95% CI: 1.15−1.85, *p* < 0.01) in the highest quartile of the uPDI compared to the lowest quartile and by 58% (OR = 1.58, 95% CI: 1.17−2.12, *p* < 0.01) in those without abdominal obesity.

Overall, the interaction between the PDI, hPDI, uPDI, and abdominal obesity in their effect on the prevalence of depression and anxiety was shown to be statistically significant. The negative multiplicative effect of having abdominal obesity and the highest quartile of PDI, hPDI, and the lowest quartile of uPDI on depression and anxiety was more pronounced compared to not having abdominal obesity (all *p* < 0.01) (Appendix A).

Additional subgroup analyses were conducted using the fully adjusted model. After dividing participants according to sex, the effects of the plant-based diet indices on anxiety remained significant for women but not for men (Appendix A). After further subgrouping of participants according to participants’ age (≤80 and >80 years old), residence, economic situation, and comorbidity, the patterns of effects of plant-based diet indices and abdominal obesity with both mental disorders were broadly analogous to the main results but were significantly attenuated in the younger group (age ≤ 80 years), urban population, economically affluent individuals, and those with chronic disease. (Appendix A).

## 4. Discussion

In this study, we discovered that a higher PDI and hPDI were linked to a lower prevalence of depression and anxiety, and a higher uPDI was linked to a higher prevalence of depression and anxiety. Abdominal obesity was linked to a lower prevalence of depression and anxiety. The association between these plant-based diet indices and depression and anxiety was largely enhanced in participants who did not have abdominal obesity. In addition, we observed a significant interaction between the plant-based diet indices and abdominal obesity in their effect on the prevalence of depression and anxiety. Except for the absence of an association between the plant-based diet indices and anxiety in men, the results of the subgroup analysis were generally consistent with the primary results.

We found that higher consumption of plant-based foods, especially healthful plant-based foods, may be able to reduce the prevalence of depression and anxiety in participants, which is largely aligned with previous epidemiological findings. An observational retrospective study in Spain showed that people who did not have depression tended to consume more legumes, fruits, and vegetables than those with depression [13]. A study conducted among African Americans demonstrated a protective effect of vegetable intake on clinically relevant levels of depressive symptoms by combining cross-sectional and cohort studies [45]. A cohort study based on a Chinese population concluded that frequent green tea consumption is associated with a decreased risk of depressive symptoms [46]. We further observed that greater consumption of animal-based foods was associated with a higher prevalence of depression and anxiety. Several former pieces of research have yielded similar results. A cross-sectional research study using UK Biobank data showed an adverse association between the intake of processed meat and milk and mental health [47]. Another cohort study based on a Chinese population also showed that no animal-based food (meat and fish) intake was associated with a lower risk of depression compared to those who consumed animal-based foods [48]. The following pathophysiological mechanisms could help to corroborate our findings. First, animal studies and clinical observational trials indicate that inflammation and oxidative stress may trigger the pathogenesis of depression and anxiety by affecting brain function [49,50]. Plant-based foods are often rich in anti-inflammatory and anti-oxidative flavonoids and antioxidants, which can reduce inflammation and oxidative stress and can be effective in forestalling the progression of mental disorders [51,52]. Second, results from mice tests and clinical randomized controlled trials suggest that deficiencies of zinc and magnesium are not only associated with significant increases in inflammatory markers [53,54] but also have an impact on the function of the hypothalamic–pituitary–adrenal (HPA) axis, which exerts an essential effect in mental disorders such as depression [55,56,57]. As a good source of these micronutrients, plant-based foods have potential antidepressant and anxiolytic activity. Finally, a high intake of animal-based foods, including meat, sugar, and dairy products, is often related to a higher risk of low-grade inflammation and cardiovascular disease, which are both linked to the pathogenesis of depression and anxiety [58,59].

We found that having abdominal obesity may be related to a lower prevalence of depression and anxiety. The connection between obesity and mental disorders has been explored in several previous studies, but the conclusions remain contentious. Partial studies have suggested that abdominal obesity is associated with an increased risk of depression and anxiety [30,60,61,62]. In contrast, other studies that were also conducted among Chinese populations have suggested a protective effect of abdominal obesity against mental disorders [31,35]. In addition, some studies have suggested that the association between abdominal obesity and mental disorders varies by gender [62,63]. The “happy fat” hypothesis proposed by Crisp et al. [64] could explain the inverse relationship between abdominal obesity and the occurrence of mental disorders in this study. This hypothesis proposes that people who maintain their weight by restricting their diet are more at risk of increased depression owing to food deprivation and, therefore, obese people may be less susceptible to strict diets that can lead to depression [64]. Furthermore, higher levels of obesity may lead to increased insulin resistance and increased serum-free fatty acids and tryptophan cycling, which subsequently affects HPA axis function [65,66,67]. Contrary to the stigmatization of obesity in the West [68], Chinese people tend to have a more positive attitude towards obesity, which is associated with the fact that in traditional Chinese culture, obesity represents high socioeconomic status and good morals [33,69].

We first demonstrated that the correlation between plant-based diet indices and depression and anxiety is largely enhanced in individuals without abdominal obesity. We further verified the interaction between abdominal obesity and plant-based diet indices through interaction analysis. In colloquial terms, the mental health and physical health problems associated with obesity may lead people to make a conscious effort to control their weight, and controlling their diet often has an important role in obesity management [70]. In addition, adopting weight loss diets, such as low-fat and anti-inflammatory diets, lead people to consume more healthful plant-based foods [71,72,73], and maintaining a healthy diet reduces the risk of obesity [74]. A previous study found that people with a genetic predisposition to higher BMI may have a more stable psychological state according to phenomic scan analysis [75]. This may explain why the protective effects of a high PDI and high hPDI on depression and anxiety and the deleterious effects of a high uPDI on depression and anxiety are more pronounced in non-abdominally obese people; however, the exact mechanisms remain to be explored.

To our knowledge, there are no previous studies investigating the relationship between the three plant-based diet indices and mental disorders in Chinese populations, so this study validates their association. The results of this study have good generalization implications due to the national representativeness of the data used. In addition, we performed a comprehensive adjustment for covariates and conducted subgroup analyses, allowing us to minimize the interference of potential confounders with the findings. There are also some limitations of this study. First, because this was a cross-sectional study, inferences of causality cannot be made. Second, the data were self-reported by participants, which may lead to recall bias. Third, the plant-based diet indices do not take into account the degree of food processing in the construction process. Therefore, the plant-based diet indices have limitations in evaluating the effect of food processing degree on anxiety and depression. Fourth, the simplified food frequency questionnaire in the CLHLS only collects the frequency of food consumption, so we cannot calculate and adjust the total energy intake. Additionally, the food categories included in the simplified food frequency questionnaire are relatively limited and may not completely cover all food categories. However, a large number of studies have previously demonstrated the reliability and validity of the simplified food frequency questionnaire, and its efficacy has been widely recognized. Fifth, the two scales used in this study could not be used to define the severity of depression and anxiety, which may lead to bias. Sixth, the CLHLS primarily includes participants aged 65 years and older, which may have led to a lack of age representation in the results of this study. Finally, since the participants included in this study were all from China, the specificity of both dietary habits and WC judgment criteria limits the generalization of the findings to other general groups and ethnicities.

## 5. Conclusions

This large study based on older Chinese adults showed that a higher PDI and hPDI were associated with a lower prevalence of depression and anxiety, and a higher uPDI was associated with a higher prevalence of depression and anxiety. In addition, abdominal obesity was related to a lower prevalence of depression and anxiety. There was a significant interaction between these plant-based diet indices and abdominal obesity in their effect on the prevalence of depression and anxiety. The association between these plant-based diet indices and depression and anxiety was largely enhanced in participants who did not have abdominal obesity. These results support the beneficial effect of increasing the intake of plant-based foods and decreasing the intake of animal-based foods. Furthermore, abdominal obesity is inextricably linked to eating habits, and further validation of the potential role and mechanisms of abdominal obesity in the association between eating and mental disorders in cohort studies or intervention trials is warranted.

## Figures and Tables

**Figure 1 nutrients-15-02721-f001:**
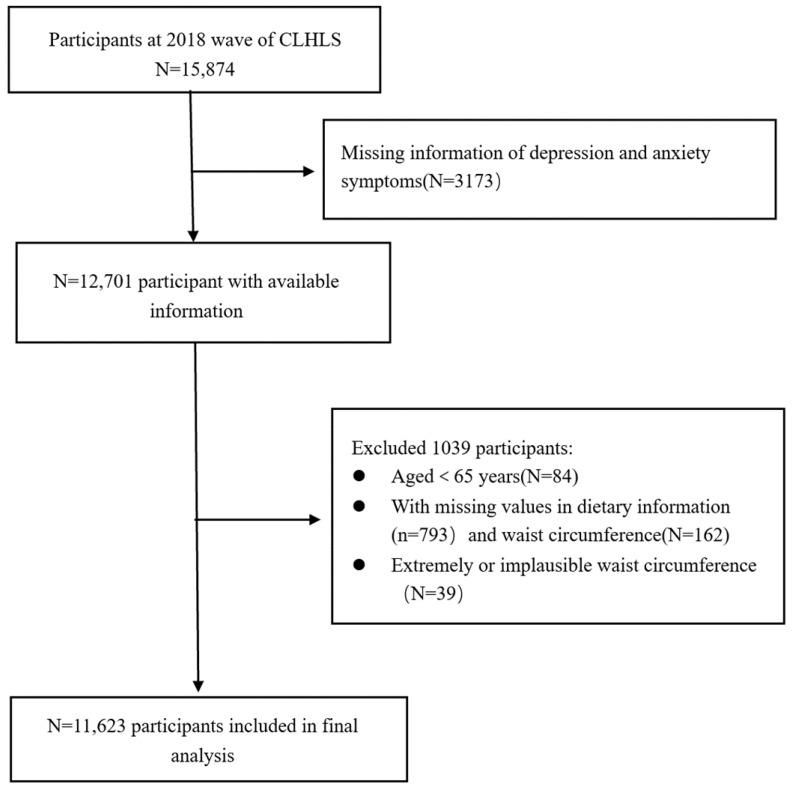
Flowchart of participant screening flowchart.

**Figure 2 nutrients-15-02721-f002:**
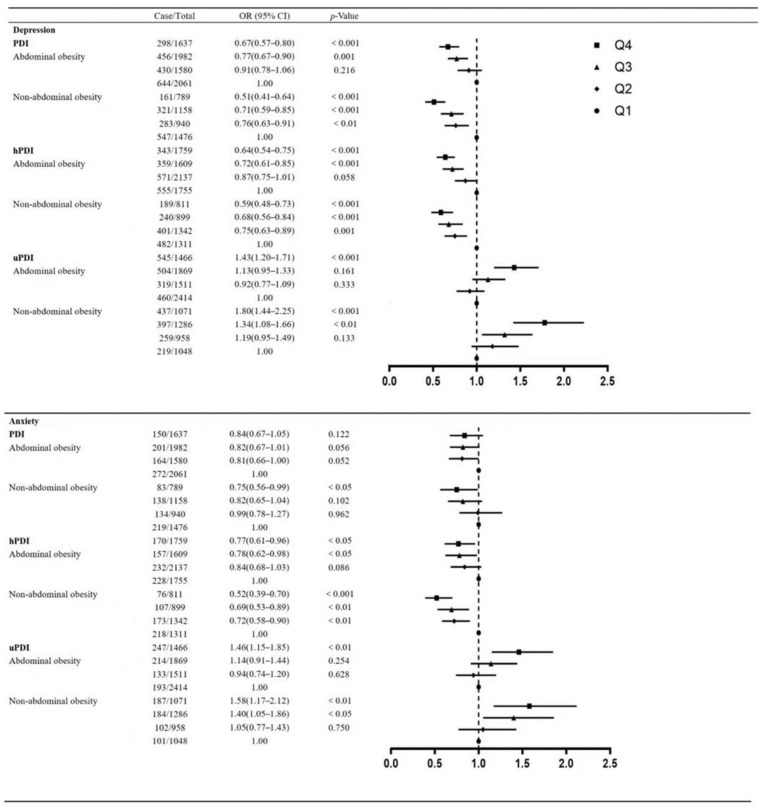
ORs and 95% CIs for developing depression and anxiety by plant-based diet indices in the fully adjusted model, stratified by abdominal obesity.

**Table 1 nutrients-15-02721-t001:** Comparison of the prevalence of depression and anxiety among participants with different characteristics.

Characteristics	Total	Non-Depressed	Depressed	*p*-Value	Non-Anxiety	Anxiety	*p*-Value
N	11,623	8483 (73.0)	3140 (27.0)		10,262 (88.3)	1361 (11.7)	
PDI score	48.11 ± 5.50	48.55 ± 5.45	46.95 ± 5.47	<0.001	48.22 ± 5.49	47.32 ± 5.48	<0.001
hPDI score	46.66 ± 5.37	47.02 ± 5.33	45.68 ± 5.37	<0.001	46.77 ± 5.33	45.79 ± 5.59	<0.001
uPDI score	49.13 ± 6.92	48.44 ± 6.84	51.00 ± 6.79	<0.001	48.86 ± 6.87	51.15 ± 6.97	<0.001
Abdominal obesity	7260 (62.5)	5432 (64.0)	1828 (58.2)	<0.001	6473 (63.1)	787 (57.8)	<0.001
Age, years	83.21 ± 10.98	82.72 ± 11.04	84.54 ± 10.71	<0.001	83.26 ± 10.98	82.87 ± 11.01	0.188
Sex, female	6197 (53.3)	4270 (50.3)	1927 (61.4)	<0.001	5318 (51.8)	879 (64.6)	<0.001
Residence				<0.001			<0.001
Urban	2743 (23.6)	2128 (25.1)	615 (19.6)		2498 (24.3)	245 (18.0)	
Town	8880 (76.4)	6355 (74.9)	2525 (80.4)		7764 (75.7)	1116 (82.0)	
Marital status				<0.001			<0.01
Married/cohabitating	5256 (45.2)	4090 (48.2)	1166 (37.1)		4689 (45.7)	567 (41.7)	
Others	6367 (54.8)	4393 (51.8)	1974 (62.9)		5573 (54.3)	794 (58.3)	
Cohabitation status				<0.001			0.010
Solitude	1946 (16.7)	1280 (15.1)	666 (21.2)		1685 (16.4)	261 (19.2)	
Not living alone	9677 (83.3)	7203 (84.9)	2474 (78.8)		8577 (83.6)	1100 (80.8)	
Education				<0.001			<0.001
Illiterate	5309 (45.7)	3521 (41.5)	1788 (56.9)		4545 (44.3)	764 (56.1)	
Primary	3927 (33.8)	3041 (35.8)	886 (28.2)		3533 (34.4)	394 (28.9)	
Secondary and above	2387 (20.5)	1921 (22.6)	466 (14.8)		2184 (21.3)	203 (14.9)	
Occupation				<0.001			0.001
Agricultural	7287 (62.7)	5156 (60.8)	2131 (67.9)		6380 (62.2)	907 (66.6)	
Others	4336 (37.3)	3327 (39.2)	1009 (32.1)		3882 (37.8)	454 (33.4)	
Economic situation				<0.001			<0.001
Wealthy	2424 (20.9)	2051 (24.2)	373 (11.9)		2265 (22.1)	159 (11.7)	
Not wealthy	9199 (79.1)	6432 (75.8)	2767 (88.1)		7997 (77.9)	1202 (88.3)	
Sleep duration				<0.001			<0.001
≤6 h	4398 (37.8)	2678 (31.6)	1720 (54.8)		3616 (35.2)	782 (57.5)	
7–8 h	4345 (37.4)	3495 (41.2)	850 (27.1)		4002 (39.0)	343 (25.2)	
≥9 h	2880 (24.8)	2310 (27.2)	570 (18.2)		2644 (25.8)	236 (17.3)	
Smoking status				<0.001			<0.001
Never	7977 (68.6)	5656 (66.7)	2321 (73.9)		6955 (67.8)	1022 (75.1)	
Former	1761 (15.2)	1364 (16.1)	397 (12.6)		1595 (15.5)	166 (12.2)	
Current	1885 (16.2)	1463 (17.2)	422 (13.4)		1712 (16.7)	173 (12.7)	
Alcohol consumption				<0.001			<0.001
Never	8467 (72.8)	6052 (71.3)	2415 (76.9)		7410 (72.2)	1057 (77.7)	
Former	1367 (11.8)	1002 (11.8)	365 (11.6)		1209 (11.8)	158 (11.6)	
Current	1789 (15.4)	1429 (16.8)	360 (11.5)		1643 (16.0)	146 (10.7)	
Physical exercise				<0.001			<0.001
Yes	4049 (34.8)	3339 (39.4)	710 (22.6)		3658 (35.6)	391 (28.7)	
No	7574 (65.2)	5144 (60.6)	2430 (77.4)		6604 (64.4)	970 (71.3)	
BMI (kg/m^2^)				<0.001			<0.001
Underweight	1720 (14.8)	1121 (13.2)	599 (19.1)		1465 (14.3)	255 (18.7)	
Normal	5985 (51.5)	4365 (51.5)	1620 (51.6)		5297 (51.6)	688 (50.6)	
Overweight	2901 (25.0)	2231 (26.3)	670 (21.3)		2606 (25.4)	295 (21.7)	
Obese	1017 (8.7)	766 (9.0)	251 (8.0)		894 (8.7)	123 (9.0)	
Chronic disease				<0.001			<0.001
Yes	7005 (60.3)	4951 (58.4)	2054 (65.4)		6101 (59.5)	904 (66.4)	
No	4618 (39.7)	3532 (41.6)	1086 (34.6)		4161 (40.5)	457 (33.6)	
Comorbidity				<0.001			<0.001
Yes	2947 (25.4)	2009 (23.7)	938 (29.9)		2534 (24.7)	413 (30.3)	
No	8676 (74.6)	6474 (76.3)	2202 (70.1)		7728 (75.3)	948 (69.7)	
Sedentary leisure activities				<0.001			<0.001
Yes	7446 (64.1)	5839 (68.8)	1607 (51.2)		6762 (65.9)	684 (50.3)	
No	4177 (35.9)	2644 (31.2)	1533 (48.8)		3500 (34.1)	677 (49.7)	
Active leisure activities				<0.001			<0.05
Yes	3717 (32.0)	2939 (34.6)	778 (24.8)		3316 (32.3)	401 (29.5)	
No	7906 (68.0)	5544 (65.4)	2362 (75.2)		6946 (67.7)	960 (70.5)	

PDI—overall plant-based diet index; hPDI—healthful plant-based diet index; uPDI—unhealthful plant-based diet index; BMI—body mass index.

**Table 2 nutrients-15-02721-t002:** Association of plant-based diet indices with depression.

	Q1	Q2	Q3	Q4	*p* for Trend
		OR (95% CI)	OR (95% CI)	OR (95% CI)	
PDI					
Cases/total	1191/3537	713/2520	777/3140	459/2426	
Model 1 ^a^	1.00	0.81 (0.72–0.90)	0.69 (0.62–0.77)	0.51 (0.45–0.58)	<0.001
Model 2 ^b^	1.00	0.84 (0.75–0.94)	0.73 (0.65–0.81)	0.55 (0.48–0.62)	<0.001
Model 3 ^c^	1.00	0.85 (0.76–0.96)	0.75 (0.67–0.85)	0.61 (0.54–0.70)	<0.001
hPDI					
Cases/total	1037/3066	972/3479	599/2508	532/2570	
Model 1	1.00	0.80 (0.72–0.89)	0.67 (0.59–0.75)	0.57 (0.50–0.64)	<0.001
Model 2	1.00	0.83 (0.74–0.92)	0.71 (0.63–0.80)	0.60 (0.53–0.68)	<0.001
Model 3	1.00	0.82 (0.73–0.91)	0.71 (0.63–0.81)	0.62 (0.54–0.71)	<0.001
uPDI					
Cases/total	679/3462	578/2469	901/3155	982/2537	
Model 1	1.00	1.18 (1.04–1.34)	1.51 (1.33–1.70)	2.30 (2.03–2.60)	<0.001
Model 2	1.00	1.10 (0.97–1.26)	1.35 (1.19–1.53)	1.94 (1.71–2.21)	<0.001
Model 3	1.00	1.02 (0.89–1.16)	1.19 (1.04–1.36)	1.55 (1.35–1.78)	<0.001

OR—odds ratio; CI—confidence interval. Grouping basis of three plant-based diet indices: PDI (Q1 ≤ 45, 46 ≤ Q2 ≤ 48, 49 ≤ Q3 ≤ 52, Q4 ≥ 53); hPDI (Q1 ≤ 43, 44 ≤ Q2 ≤ 47, 48 ≤ Q3 ≤ 50, Q4 ≥ 51); uPDI (Q1 ≤ 45, 46 ≤ Q2 ≤ 49, 50 ≤ Q3 ≤ 54, Q4 ≥ 55). ^a^ Adjusted for demographic characteristics; ^b^ adjusted for demographic characteristics and socioeconomic characteristics; ^c^ adjusted for demographic characteristics, socioeconomic characteristics, health behaviors, and physical status.

**Table 3 nutrients-15-02721-t003:** Association of plant-based diet indices with anxiety.

	Q1	Q2	Q3	Q4	*p* for Trend
		OR (95% CI)	OR (95% CI)	OR (95% CI)	
PDI					
Cases/total	491/3537	298/2520	339/3140	233/2426	
Model 1 ^a^	1.00	0.84 (0.72–0.99)	0.76 (0.65–0.88)	0.69 (0.58–0.81)	<0.001
Model 2 ^b^	1.00	0.87 (0.74–1.02)	0.79 (0.68–0.92)	0.73 (0.61–0.86)	0.001
Model 3 ^c^	1.00	0.89 (0.76–1.05)	0.83 (0.71–0.97)	0.81 (0.68–0.96)	<0.05
hPDI					
Cases/total	446/3066	405/3479	264/2508	246/2570	
Model 1	1.00	0.78 (0.67–0.90)	0.71 (0.60–0.83)	0.63 (0.54–0.75)	<0.001
Model 2	1.00	0.80 (0.69–0.92)	0.74 (0.63–0.87)	0.66 (0.56–0.78)	<0.001
Model 3	1.00	0.78 (0.67–0.91)	0.74 (0.63–0.88)	0.66 (0.56–0.79)	<0.001
uPDI					
Cases/total	294/3462	235/2469	398/3155	434/2537	
Model 1	1.00	1.08 (0.90–1.30)	1.44 (1.22–1.71)	2.00 (1.68–2.37)	<0.001
Model 2	1.00	1.03 (0.86–1.24)	1.35 (1.14–1.61)	1.80 (1.51–2.15)	<0.001
Model 3	1.00	0.99 (0.82–1.20)	1.25 (1.04–1.49)	1.50 (1.25–1.80)	<0.001

^a^ Adjusted for demographic characteristics; ^b^ adjusted for demographic characteristics and socioeconomic characteristics; ^c^ adjusted for demographic characteristics, socioeconomic characteristics, health behaviors, and physical status.

**Table 4 nutrients-15-02721-t004:** Association of abdominal obesity with depression and anxiety.

	Non- Abdominal Obesity	Abdominal Obesity	
		OR (95% CI)	*p*-Value
Depression			
Cases/total	1312/4363	1828/7260	
Model 1 ^a^	1.00	0.79 (0.72–0.86)	<0.001
Model 2 ^b^	1.00	0.81 (0.74–0.88)	<0.001
Model 3 ^c^	1.00	0.79 (0.72–0.87)	<0.001
Anxiety			
Cases/total	574/4363	787/7260	
Model 1	1.00	0.75 (0.67–0.84)	<0.001
Model 2	1.00	0.77 (0.68–0.86)	<0.001
Model 3	1.00	0.75 (0.66–0.85)	<0.001

^a^ Adjusted for demographic characteristics; ^b^ adjusted for demographic characteristics and socioeconomic characteristics; ^c^ adjusted for demographic characteristics, socioeconomic characteristics, health behaviors, and physical status (except for BMI).

## Data Availability

The original data used for this study can be obtained from the CLHLS (https://opendata.pku.edu.cn/ accessed on 12 April 2023).

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
