# Peer review of "Association of Plant-Based Diet Indices and Abdominal Obesity with Mental Disorders among Older Chinese Adults"

_nutrients, 2023, doi:10.3390/nu15122721_

Round 1
Reviewer 1 Report
The paper addresses a very interesting topic, which would be appreciated by the readers of Nutrients.
Despite the transversal design, the authors looked at the major aspect related to the potential relationship between the dietary habits and the anxiety/depression.
Appropriate questionnaires have been used.
Some points to address:
- the comorbidities of the patients could have been of interest
- the validity of the questionnaires for the Chinese language should have been mentioned
- the influence of the particularities of the dietary habbits on the outcome should be mentioned as a major limitation of wide applicability.
Author Response
May, 27, 2023
Manuscript ID nutrients-2368982, “Association of Plant-Based Diet Indices and Abdominal Obesity with Mental Disorders Among Chinese Older Adults”
Dear editor and reviewers:
We are very grateful to you for giving us the opportunity to revise and resubmit our manuscript. We appreciate reviewers for their valuable and constructive comments on our manuscript, which helped us to improve our manuscript.
All comments and suggestions have been carefully addressed in the revised manuscript. Detailed, point-by-point responses to each comment are included in the following pages (please note that the reviewers’ comments are in bold, and our responses are underneath the comments).
We hope the revisions meet the requirements of reviewers as well as editors. We especially look forward to the revised manuscript being accepted by the publication.
Response to reviewers' comments:
Reviewer#1: The paper addresses a very interesting topic, which would be appreciated by the readers of Nutrients.
Despite the transversal design, the authors looked at the major aspect related to the potential relationship between the dietary habits and the anxiety/depression.
Appropriate questionnaires have been used.
Reply:
Thank you very much for reviewing our paper. We appreciate your comments and admire your rigorous attitude towards research. Your patient guidance will not only make this article more scientific but also guide our future works. After careful discussion of the comments, we have substantially revised the manuscript.
Major comments:
- The comorbidities of the patients could have been of interest.
Reply:
Thank you for your valuable comments. After some discussion, perhaps the “comorbidities” you mentioned refer to the chronic disease “comorbidity” that was adjusted for as a covariate in our manuscript. Compared to previous studies, we believe that comorbidity may be a physical condition that affects depression and anxiety in older adults, and based on the performance of the data in the model, we found that adjusting for comorbidity did improve the accuracy of the model. Therefore, we further conducted subgroup analyses according to the status of participants' comorbidities as you suggested. The results showed a broadly similar pattern of effects for the combined two mental disorders of the vegetative diet index and abdominal obesity as the primary outcome, but significantly attenuated in patients with chronic disease. We have summarized and presented this result in the manuscript.
If the “comorbidities” and results described here are not what you would like to know, please do not hesitate to send us your valuable comments.
Changes in the revised manuscript:
Page 11, paragraph 2, line 4: “After further subgrouping of participants according to participants’ age (≤80 and >80 years old), residence, economic situation, and comorbidity, the patterns of effects of plant-based diet indices and abdominal obesity with both mental disorders were broadly analogous to the main results but were significantly attenuated in the young-er group (age ≤ 80 years), urban population, economically affluent individuals, and those with chronic disease. (Tables S6−S13). ”
Table S12. Association of three plant-based diet indices with depression and anxiety, stratified by comorbidity.
|
  |
Q1 |
  |
Q2 |
  |
Q3 |
  |
Q4 |
||||||||||||
|
  |
Case/total |
OR (95% CI) |
  |
Case/total |
OR (95% CI) |
p-Value |
  |
Case/total |
OR (95% CI) |
p-Value |
  |
Case/total |
OR (95% CI) |
p-Value |
|||||
|
Depression |
|||||||||||||||||||
|
Comorbidity |
|||||||||||||||||||
|
PDI |
281/759 |
1.00 |
226/674 |
0.96(0.76-1.22) |
0.758 |
256/846 |
0.92(0.73-1.15) |
0.453 |
175/668 |
0.82(0.64-1.06) |
0.126 |
||||||||
|
hPDI |
239/617 |
1.00 |
305/899 |
0.92(0.73-1.16) |
0.499 |
197/690 |
0.78(0.60-1.00) |
<0.05 |
197/741 |
0.76(0.59-0.98) |
<0.05 |
||||||||
|
uPDI |
317/1169 |
1.00 |
195/645 |
0.93(0.73-1.17) |
0.521 |
193/607 |
0.91(0.71-1.17) |
0.459 |
233/526 |
1.31(1.01-1.70) |
<0.05 |
||||||||
|
Non-comorbidity |
|||||||||||||||||||
|
PDI |
910/2778 |
1.00 |
487/1846 |
0.83(0.72-0.95) |
<0.01 |
521/2294 |
0.70(0.61-0.80) |
<0.001 |
284/1758 |
0.55(0.47-0.65) |
<0.001 |
||||||||
|
hPDI |
798/2449 |
1.00 |
667/2580 |
0.79(0.69-0.90) |
<0.001 |
402/1818 |
0.70(0.61-0.82) |
<0.001 |
335/1829 |
0.59(0.50-0.68) |
<0.001 |
||||||||
|
uPDI |
362/2293 |
1.00 |
383/1824 |
1.08(0.91-1.28) |
0.375 |
708/2548 |
1.33(1.13-1.55) |
<0.001 |
749/2011 |
1.70(1.44-2.01) |
<0.001 |
||||||||
|
Anxiety |
|||||||||||||||||||
|
Comorbidity |
|||||||||||||||||||
|
PDI |
120/759 |
1.00 |
101/674 |
1.05(0.78-1.42) |
0.746 |
114/846 |
0.98(0.73-1.32) |
0.894 |
78/668 |
0.88(0.64-1.22) |
0.440 |
||||||||
|
hPDI |
99/617 |
1.00 |
127/899 |
0.89(0.66-1.20) |
0.427 |
90/690 |
0.87(0.63-1.20) |
0.384 |
97/741 |
0.87(0.63-1.20) |
0.395 |
||||||||
|
uPDI |
114/1169 |
1.00 |
84/645 |
1.15(0.84-1.59) |
0.378 |
96/607 |
1.30(0.94-1.80) |
0.109 |
119/526 |
1.66(1.19-2.31) |
<0.01 |
||||||||
|
Non-comorbidity |
|||||||||||||||||||
|
PDI |
371/2778 |
1.00 |
197/1846 |
0.84(0.70-1.01) |
0.069 |
225/2294 |
0.78(0.65-0.94) |
<0.01 |
155/1758 |
0.79(0.64-0.98) |
<0.05 |
||||||||
|
hPDI |
347/2449 |
1.00 |
278/2580 |
0.75(0.63-0.90) |
0.001 |
174/1818 |
0.70(0.57-0.86) |
0.001 |
149/1829 |
0.59(0.48-0.73) |
<0.001 |
||||||||
|
uPDI |
180/2293 |
1.00 |
  |
151/1824 |
0.91(0.72-1.16) |
0.458 |
  |
302/2548 |
1.20(0.97-1.49) |
0.092 |
  |
315/2011 |
1.42(1.14-1.77) |
<0.01 |
|||||
Estimates are odds ratios (95% confidence intervals) from multivariable logistic regression models adjusted for age, sex, residence, marital status, cohabitation status, education, occupation, economic situation, sleep duration, smoking status, alcohol consumption, physical exercise, leisure activity participation status, BMI and chronic disease.
Table S13. Association of abdominal obesity with depression and anxiety, stratified by comorbidity.
|
  |
Non-abdominal obesity |
  |
Abdominal obesity |
||||
|
  |
Case/total |
OR (95% CI) |
  |
Case/total |
OR (95% CI) |
p-Value |
|
|
Comorbidity |
|||||||
|
Depression |
303/762 |
1.00 |
635/2185 |
0.67(0.56-0.81) |
<0.001 |
||
|
Anxiety |
134/762 |
1.00 |
279/2185 |
0.67(0.53-0.85) |
0.001 |
||
|
Non-comorbidity |
|||||||
|
Depression |
1009/3601 |
1.00 |
1193/5075 |
0.84(0.76-0.93) |
0.001 |
||
|
Anxiety |
440/3601 |
1.00 |
  |
508/5075 |
0.78(0.67-0.90) |
<0.001 |
|
Estimates are odds ratios (95% confidence intervals) from multivariable logistic regression models adjusted for age, sex, residence, marital status, cohabitation status, education, occupation, economic situation, sleep duration, smoking status, alcohol consumption, physical exercise, leisure activity participation status and chronic disease.
- The validity of the questionnaires for the Chinese language should have been mentioned.
Reply:
Thank you for your suggestions to help us make the article more complete. Scales with high validity need to ensure that different researchers have a consistent understanding of the meaning and connotation of a research variable. The quality of the data collected by the questionnaires for the Chinese language used in the CLHLS project has been evaluated and generally accepted and applied by scholars both at home and abroad, and has become an internationally and nationally recognized cross-disciplinary research project with very rich data information and great research potential.
The depression scale (CES-D-10) and the generalized anxiety scale (GAD-7) used in this paper have been widely used in previous studies and have good content validity; we also calculated the KMO coefficients of the two scales to verify their tested structural validity. The KMO coefficients of the depression scale and the generalized anxiety scale were tested to be 0.871 and 0.927, respectively, demonstrating that they have good structural validity. We additionally calculated the Cronbach-α coefficients for the depression scale and the generalized anxiety scale, which were 0.805 and 0.918, respectively, proving that both scales have good reliability and high application value. According to your suggestion, we have added the relevant descriptions and marked them in the manuscript.
Changes in the revised manuscript:
Page 4, paragraph 4, line 5: “Previously, the questionnaires for the Chinese language used in CLHLS has been widely accepted and used[31,43,44]. In this paper, the Cronbach-α coefficients for the CES-D-10 and the GAD-7 were 0.805 and 0.918, respectively, and the KMO coefficients were 0.871 and 0.927, respectively, indicating good reliability and validity of the scales. ”
- The influence of the particularities of the dietary habbits on the outcome should be mentioned as a major limitation of wide applicability.
Reply:
Thank you for pointing out our problems. As you suggested, since this thesis is a study based on a group of elderly Chinese, the dietary habits of the study subjects are inevitably specific and therefore cannot represent universal dietary habits. This leads to limitations in the generalizability of the findings. Therefore, we added this limitation in the discussion section of the manuscript.
Changes in the revised manuscript:
Page 14, paragraph 1, line 1: “Finally, since the participants included in this study were all from China, the specificity of both dietary habits and WC judgment criteria limits the generalization of the findings to other general groups and ethnicities.

Reviewer 2 Report
Thank you for the opportunity to review this potentially interesting manuscript.
The authors intention is to examine the association of plant-based diet indices with depression; plant-based diet indices with anxiety; and abdominal obesity with depression, and abdominal obesity with anxiety in cross-sectional measurement of elderly Chinese.
As the authors correctly point out in the limitations to the study, the cross-sectional nature means inferences of causality cannot be made. This is an important principle. The use of the word ‘risk’ (eg ‘factor A was linked to an increased risk….’) implies causality because in common usage, risk refers to a future event. ‘Risk’ might be replaced with ‘prevalence’ in the conclusions and where relevant in the discussion. The concluding suggestion that the results of this study can help in developing reasonable dietary recommendations based on obesity status to prevent depression and anxiety in elderly populations is inappropriate unless the authors are making inferences of causality. A conclusion that randomised trials be conducted to determine whether ….. seems more appropriate. Many of the studies cited are also cross-sectional and the authors should take particular care to identify these – it would seem that clinical trials or cohort studies addressing their central interest are uncommon.
Little information is given on how subjects were selected for the CLHLS or what the response rate was.
I think it is unusual to control for BMI status when one of your explanatory variables is abdominal obesity status – I would expect abdominal obesity to be closely correlated with BMI status. However the effect estimates do not change greatly between model 2 and model 3 in Table 4 – could the authors comment on the reasons to include BMI status in model 3 and any implication of these results?
The dietary measurement uses a food frequency approach for a relatively small number of food groups, however the authors have been able to separate the foods into plant-based (healthy and unhealthy), and animal-based food consumption, with a very simple scoring system dependent on frequency of consumption. In many Western countries this would be difficult to do because of the large consumption of mixed processed foods, but this might also be the case in urban areas of China. Could the authors comment in the discussion on the construction of the PDI – does it work well for the elderly in China because they consume minimally processed foods, or is a large number of foods consumed but not measured because they aren’t in one of the 16 food groups?
There is no statement of ethics review and approval for the study.
Minor:
Section 2.1: sentence 1 - a hypothesis has not been stated.
Sentence 3 – the sample size is small relative to the population of China and population representation is difficult to assess without information on sample selection. The sentence could be simplified by stating ‘The CLHLS covers 23 provinces in China, containing 85% of the Chinese population.’
Sentence 7 – it is incorrect to say that participants were enrolled in the study (they were enrolled in the original CLHLS). It is better to say that they were ‘included’ in this study. (also first sentence of the results.
Section 2.2 - it should be stated explicitly that this measurement was by the participant (not a study investigator, for example).
Section 2.3 – the supplementary table is very useful to understand the construction of the indices. However, I don’t understand the phrase ‘unhealthful plant-based foods were scored inversely’ and ‘healthful plant-based foods were scored inversely’. Does this mean that intake of healthful plant-based foods contributed negatively to the uPDI, and intake of unhealthful plant-based food contributed negatively to the hPDI?
Section 2.6 sentence 1 – the use of the term ‘baseline’ is a little confusing because this is a cross-sectional study. This should be deleted. It is not clear what ‘differences are being examined – is this difference in characteristics by depression and anxiety status (i.e. the results in Table1)?
Sentence 5 – rather than ‘on the basis of Model 1’ perhaps ‘in addition to the covariates in model 1’.
Section 3 Results: please include a statement of the number of participants with both depression and anxiety.
Page 8, 2nd paragraph, sentence 1 – the population was stratified according to abdominal obesity status, rather than obesity status. The term ‘adherence levels’ is incorrect since participants weren’t allocated diets to which they were expected to comply. This might be phrased as …the association of quartile membership of PDI, hPDI and uPDI with depression and anxiety was determined (Figure 2). The following sentence might be ‘Specifically, the odds ratio for depression was lower by 32% (OR 0.68, 95% CI:0.58-0.81, p<0.001) in the highest quartile of the PDI in participants without abdominal obesity compared to the lowest quartile, and…’
The title for Figure 2 should note that results for the fully adjusted model (i.e. Model 3) are presented. It is not clear why the results are listed vertically from Quartile 4 to Quartile 1, while the legend is listed vertically from Q1 to Q4.
The interaction between the PDI, HPDI, uPDI and abdominal obesity for the outcomes of depression and anxiety is stated to be statistically significant (below Figure 2). Because this applies to 6 different models, the statement should be (all p<0.05, data not shown). The data in tables S2-S3 are relevant, but the statement of statistical significance for the interaction is a modelled result.
Discussion:
The authors should review the language used to avoid making inferences of causality (which the study design can’t support). For example, the first sentence of the second paragraph is a statement of causality.
The authors suggest that the difference in results by gender were small however this might be better phrased in the discussion as ‘except for the absence of an association between plant-based diet indices and anxiety in men’.
The statement about association of depression and anxiety with greater consumption of animal based food is not anchored to a reference and may be a statement from analysis done as part of the present study. However, this data is not shown and the statement isn’t particularly need and should be removed.
Where cited study findings are based on animals (mice), this should be stated
The quality of English expression is generally very high. In a few places, word choice is not precise and meaning is not clear but this occurs in only a handful of places.
Author Response
Thank you for the opportunity to review this potentially interesting manuscript.
The authors intention is to examine the association of plant-based diet indices with depression; plant-based diet indices with anxiety; and abdominal obesity with depression, and abdominal obesity with anxiety in cross-sectional measurement of elderly Chinese.
Reply:
Thank you very much for reviewing our paper. We appreciate your comments and admire your rigorous attitude towards research. Your patient guidance will not only make this article more scientific but also guide our future works. After careful discussion of the comments, we have substantially revised the manuscript.
Major comments:
Point 1. As the authors correctly point out in the limitations to the study, the cross-sectional nature means inferences of causality cannot be made. This is an important principle. The use of the word ‘risk’ (eg ‘factor A was linked to an increased risk….’) implies causality because in common usage, risk refers to a future event. ‘Risk’ might be replaced with ‘prevalence’ in the conclusions and where relevant in the discussion. The concluding suggestion that the results of this study can help in developing reasonable dietary recommendations based on obesity status to prevent depression and anxiety in elderly populations is inappropriate unless the authors are making inferences of causality. A conclusion that randomised trials be conducted to determine whether ….. seems more appropriate. Many of the studies cited are also cross-sectional and the authors should take particular care to identify these – it would seem that clinical trials or cohort studies addressing their central interest are uncommon.
Reply:
- Thank you for your keen observation of the lack of vocabulary in the article and for your rigorous suggestions. We are very sorry for our wording error, and we have replaced "risk" with "prevalence" in the corresponding section of the article based on your suggestion.
Reply:
- In addition, considering our study as a cross-sectional study, our presentation of the conclusions was indeed inappropriate. Therefore, by referring to other studies of the same type, we have more tightly revised our conclusions in terms of the significance of the study.
Changes in the revised manuscript:
Page 14, paragraph 2, line 10: “These results support the beneficial effect of increasing the intake of plant-based foods and decreasing the intake of animal-based foods. Furthermore, abdominal obesity is inextricably linked to eating habits, and further validation of the potential role and mechanisms of abdominal obesity in the association between eating and mental disorders in cohort studies or intervention trials is warranted. ”
Reply:
- Finally, your comments make us aware of the limitations of the discussion section. Compared to cross-sectional studies, cohort studies and clinical trials have a superior ability to argue causality. In conjunction with your suggestions, we have added some cohort studies in the discussion section that together with the existing cross-sectional studies provide support for our views.
Changes in the revised manuscript:
Page 12, paragraph 2, line 1: “We found that a higher consumption of plant-based foods, especially healthful plant-based foods, may be able to reduce the prevalence of depression and anxiety in participants, which is largely aligned with previous epidemiological findings. An observational retrospective study in Spain showed that people who did not have depression tended to consume more legumes, fruits, and vegetables than those with depression[13]. A study conducted among African Americans demonstrated a protective effect of vegetable intake on clinically-relevant levels of depressive symptoms by combining cross-sectional and cohort studies[45]. A cohort study based on a Chinese population concluded that frequent green tea consumption is associated with a decreased risk of depressive symptoms[46]. We further observed that greater consumption of animal-based foods was associated with a greater prevalence risk of depression and anxiety. Several former pieces of research have yielded similar results. A cross-sectional research using UK Biobank data showed an adverse association between the intake of processed meat and milk and mental health[47]. Another cohort study based on a Chinese population also showed that no animal-based food (meat and fish) intake was associated with a lower risk of depression compared to those who consumed animal-based foods[48]. ”
Page 13, paragraph 1, line 2: “Partial studies have suggested that abdominal obesity is associated with an increased risk of depression and anxiety[30,60-62]. ”
Reference:
- Ribeiro S M L, Malmstrom T K, Morley J E, et al. Fruit and vegetable intake, physical activity, and depressive symptoms in the African American Health (AAH) study. J Affect Disord, 2017, 220: 31-37.
- Dong X, Gu Y, Rayamajhi S, et al. Green tea consumption and risk of depressive symptoms: Results from the TCLSIH Cohort Study. J Affect Disord, 2022, 310: 183-188.
- Shen Y C, Chang C E, Lin M N, et al. Vegetarian Diet Is Associated with Lower Risk of Depression in Taiwan. Nutrients, 2021, 13(4).
- Mulugeta A, Zhou A, Power C, et al. Obesity and depressive symptoms in mid-life: a population-based cohort study. BMC Psychiatry, 2018, 18(1): 297.
Point 2. Little information is given on how subjects were selected for the CLHLS or what the response rate was.
Reply:
We are deeply aware of the problem of insufficient information about the research subjects in CLHLS as you suggested. According to the information we have checked, CLHLS conducted face-to-face questionnaires in randomly selected areas in 23 provinces of China by multi-stage stratified sampling method, and the face-to-face questionnaires method ensured a good response rate of CLHLS. However, CLHLS does not report detailed participant response rates, so we apologize for not being able to present detailed data. Meanwhile, the CLHLS project also ensured a rough balance of all age groups and gender in each survey area by administrative unit and gender under the premise of voluntary participation. For the completeness of the "Data sources and study population" section of the article, we have made appropriate additions. Thank you for your comments, which have helped us to improve the article.
Changes in the revised manuscript:
Page 3, paragraph 1, line 5: “The CLHLS uses a multi-stage stratified sampling method and covers 23 provinces in China, accounting for approximately 85% of the Chinese population. Meanwhile, face-to-face questionnaires were administered to participants to ensure a good response rate. More detailed descriptions of the CLHLS design can be found elsewhere[37-39]. ”
Reference:
- Liu E, Feng Y, Yue Z, et al. Differences in the health behaviors of elderly individuals and influencing factors: Evidence from the Chinese Longitudinal Healthy Longevity Survey. Int J Health Plann Manage, 2019, 34(4): e1520-e1532.
- Zeng Y. Towards Deeper Research and Better Policy for Healthy Aging --Using the Unique Data of Chinese Longitudinal Healthy Longevity Survey. China Economic J, 2012, 5(2-3): 131-149.
- Zeng Y, Feng Q, Hesketh T, et al. Survival, disabilities in activities of daily living, and physical and cognitive functioning among the oldest-old in China: a cohort study. Lancet, 2017, 389(10079): 1619-1629.
Point 3. I think it is unusual to control for BMI status when one of your explanatory variables is abdominal obesity status – I would expect abdominal obesity to be closely correlated with BMI status. However the effect estimates do not change greatly between model 2 and model 3 in Table 4 – could the authors comment on the reasons to include BMI status in model 3 and any implication of these results?
Reply:
Thanks for your insightful comments and sorry for your confusion. After receiving your comments, we immediately recognized the problem and reflected on it. When including the covariates in the model, we regarded BMI as a good indicator of body fat distribution in the study subjects, but ignored the problem of co-linearity between BMI and WC, and the inertia of thinking prevented us from catching this error in time. It is possible that the other covariates adjusted by model 3 masked the role of BMI resulting in the effect estimates of model 2 and model 3 in the original Table 4 did not change much. We no longer included BMI as a covariate in model 3 in our analysis exploring the association between abdominal obesity and depression and anxiety(Table 4). The overall effect estimates appeared to be better after the modification. We also no longer adjusted for BMI in our subsequent analyses stratified by obesity status to avoid bias from over-adjustment, and the revised conclusions were essentially similar as before the adjustment. Thank you again for giving us the opportunity to make the correction.
Changes in the revised manuscript:
Page 8, paragraph 2, line 5: “After further controlling for all covariates, abdominal obesity was still associated with a lower prevalence of depression (OR = 0.79, 95% CI: 0.72-0.87, p < 0.001) and anxiety(OR = 0.75, 95% CI: 0.66-0.85, p < 0.001) when compared with participants who did not have abdominal obesity. ”
Table 4. Association of abdominal obesity with depression and anxiety.
|
  |
Non- abdominal obesity |
Abdominal obesity |
|
|
  |
OR (95% CI) |
P -Value |
|
|
Depression |
|
||
|
Cases/total |
1312/4363 |
1828/7260 |
|
|
Model 1a |
1.00 |
0.79(0.72−0.86) |
<0.001 |
|
Model 2b |
1.00 |
0.81(0.74−0.88) |
<0.001 |
|
Model 3c |
1.00 |
0.79(0.72-0.87) |
<0.001 |
|
Anxiety |
|||
|
Cases/total |
574/4363 |
787/7260 |
|
|
Model 1 |
1.00 |
0.75(0.67−0.84) |
<0.001 |
|
Model 2 |
1.00 |
0.77(0.68−0.86) |
<0.001 |
|
Model 3 |
1.00 |
0.75(0.66-0.85) |
<0.001 |
a:adjusted for demographic characteristics; b:adjusted for demographic characteristics and socioeconomic characteristics; c:adjusted for demographic characteristics, socioeconomic characteristics, health behaviors, and physical status (except for BMI).
Point 4. The dietary measurement uses a food frequency approach for a relatively small number of food groups, however the authors have been able to separate the foods into plant-based (healthy and unhealthy), and animal-based food consumption, with a very simple scoring system dependent on frequency of consumption. In many Western countries this would be difficult to do because of the large consumption of mixed processed foods, but this might also be the case in urban areas of China. Could the authors comment in the discussion on the construction of the PDI – does it work well for the elderly in China because they consume minimally processed foods, or is a large number of foods consumed but not measured because they aren’t in one of the 16 food groups?
Reply:
Thank you for your valuable suggestions. First of all, with reference to previous literature, only the potential health effects of foods and the frequency of food intake were considered in the construction of the plant-based diet index, and the degree of food processing was not considered. Therefore, the plant-based diet index has limitations in evaluating the effects of food processing degree on depression and anxiety. Secondly, the CLHLS collects dietary information using the simplified food frequency questionnaire, which only collects the frequency of each food consumed by participants through interviews and includes a limited variety of foods. The limitations of the food frequency questionnaire prevented us from calculating the total energy intake of participants and from having a broader discussion about processed foods. However, CLHLS is an internationally and nationally recognized interdisciplinary research project, and the quality of the data collected by the questionnaires used in the project is generally recognized by national and international scholars. A number of studies have been conducted using dietary information collected by the CLHLS and have demonstrated the reliability and validity of using this simplified food frequency questionnaire to assess dietary patterns. But the inability to explore processed foods in the context of their ubiquity remains an unavoidable limitation of this study. Therefore, we have added these issues to the discussion of manuscript limitations with the aim of informing related studies that utilize databases containing more comprehensive dietary information.
Changes in the revised manuscript:
Page 13, paragraph 3, line 9: “Third, the plant-based diet indices does not take into account the degree of food processing in the construction process. Therefore, the plant-based diet indices has limitations in evaluating the effect of food processing degree on anxiety and depression. Fourth, the simplified food frequency questionnaire in the CLHLS only collects the frequency of food consumption, so we cannot calculate and adjust the total energy intake. Also the food categories included in the simple food frequency scale are relatively limited and may not completely cover all food categories. However, a large number of studies have previously demonstrated the reliability and validity of the simplified food frequency questionnaire, and its efficacy has been widely recognized. ”
Point 5.There is no statement of ethics review and approval for the study.
Reply:
Thank you for your suggestions. In fact, we set out the ethical review and approval statement for the study at the end of the manuscript, but it seems to be difficult to be noticed. To make the ethical review and approval statement easier to find, we have added it in the data sources section of the manuscript.
Changes in the revised manuscript:
Page 3, paragraph 1, line 10: “The CLHLS was approved by the Biomedical Ethics Committee of Peking University, China (IRB00001052–13074), and all participants signed an informed consent.”
Minor comments:
Point 1. Section 2.1: sentence 1 - a hypothesis has not been stated.
Reply:
Thanks for your valuable suggestions. We have rephrased this part.
Changes in the revised manuscript:
Page 2, paragraph 5, line 1: “In order to explore the association of plant-based dietary index and abdominal obesity with depression and anxiety, we used cross-sectional data from the 2018 Chinese Longitudinal Healthy Lifespan Survey (CLHLS) for examination.”
Point 2. Sentence 3 – the sample size is small relative to the population of China and population representation is difficult to assess without information on sample selection. The sentence could be simplified by stating ‘The CLHLS covers 23 provinces in China, containing 85% of the Chinese population.’
Reply:
We strongly agree with your suggestions. In conjunction with your previous suggestion, we have revised this sentence.
Changes in the revised manuscript:
Page 3, paragraph 1, line 5: “The CLHLS uses a multi-stage stratified sampling method and covers 23 provinces in China, accounting for approximately 85% of the Chinese population. ”
Point 3. Sentence 7 – it is incorrect to say that participants were enrolled in the study (they were enrolled in the original CLHLS). It is better to say that they were ‘included’ in this study. (also first sentence of the results.
Reply:
We appreciate your careful suggestions and apologize for the inappropriate wording. We have made changes based on your suggestions.
Changes in the revised manuscript:
Page 3, paragraph 2, line 5: “A total of 11,623 participants were included in the analyses. ”
Point 4. Section 2.2 - it should be stated explicitly that this measurement was by the participant (not a study investigator, for example).
Reply:
Thank you for pointing out our mistakes and we are sorry for the unclear description. We have made changes to the manuscript based on your suggestions.
Changes in the revised manuscript:
Page 3, paragraph 3, line 1: “When measuring waist circumference, participants were asked not to inhale and the interviewer would record the waist circumference by applying a tape measure directly against the skin. ”
Point 5. Section 2.3 – the supplementary table is very useful to understand the construction of the indices. However, I don’t understand the phrase ‘unhealthful plant-based foods were scored inversely’ and ‘healthful plant-based foods were scored inversely’. Does this mean that intake of healthful plant-based foods contributed negatively to the uPDI, and intake of unhealthful plant-based food contributed negatively to the hPDI?
Reply:
We apologize that our description of the scoring was confusing to you. First of all, it is important to clarify that “scored inversely” do not mean that this food contributes negatively to the plant-based diet indices. As already mentioned in the manuscript, each food is scored on a scale of 1 to 5. Therefore, no food will have a negative score. For example, for the hPDI, the higher the intake of healthy plant foods, the higher the score, while the “unhealthful plant-based foods were scored inversely” means that the higher the intake of unhealthy plant foods, the lower the score (1 for most frequently consumed, 5 for rarely or never consumed). In order to prevent ambiguity, we have reviewed the relevant literature, and we have added clarification to the scoring rules of “unhealthful plant-based foods were scored inversely” in the manuscript. The additional clarification also facilitates the understanding of the “ healthful plant-based foods were scored inversely” below.
Changes in the revised manuscript:
Page 4, paragraph 2, line 9: “For the hPDI, the higher the intake of healthful plant-based foods, the higher the score; unhealthful plant-based foods were scored inversely (the higher the intake, the lower the score). ”
Point 6. Section 2.6 sentence 1 – the use of the term ‘baseline’ is a little confusing because this is a cross-sectional study. This should be deleted. It is not clear what ‘differences are being examined – is this difference in characteristics by depression and anxiety status (i.e. the results in Table1)?
Reply:
Thank you for your careful pointing out the improper wording of the manuscript. The use of "baseline" in a cross-sectional study is indeed inappropriate, so we have removed this word. However, since this sentence is used to summarize the statistical methods used to calculate the information in Table 1, and it is still necessary to retain this section in statistical methods, so we have revised it. In addition, the purpose of Table 1 is to compare differences in the prevalence of depression and anxiety among participants with different characteristics. If the differences are statistically significant, it indicates that the prevalence of depression and anxiety may be influenced by different characteristics such as age, gender, or place of residence. The purpose of calculating the data in Table 1 is to present the reader with the current status of the prevalence of depression and anxiety.
Changes in the revised manuscript:
Page 5, paragraph 3, line 1: “Descriptive statistics were used to compare differences in the prevalence of depression and anxiety in participants with different characteristics. ”
Point 7. Sentence 5 – rather than ‘on the basis of Model 1’ perhaps ‘in addition to the covariates in model 1’.
Reply:
Since we did not express it clearly, we are sorry for your misunderstanding. In fact, what we mean exactly is that model 1controlled for participants' demographic characteristics, and model 2 adjusted not only for participants' demographic characteristics but also for their socioeconomic characteristics. The three models reduce the bias of confounders and increase the scientific validity of the results by incorporating new covariates. By referring to other authoritative studies,We have modified the corresponding parts in the description of variables adjusted by different models to make the expression more accurate and clearer.
Changes in the revised manuscript:
Page 5, paragraph 3, line 9: “Model 1 controlled for participants' demographic characteristics (age, sex, and residence). Model 2 further adjusted for socioeconomic characteristics (marital status, co-habitation status, education, occupation, and economic situation). ”
Point 8. Section 3 Results: please include a statement of the number of participants with both depression and anxiety.
Reply:
Thanks for your valuable suggestions. We have added relevant descriptions in the manuscript.
Changes in the revised manuscript:
Page 6, paragraph 1, line 4: “Moreover, 983 (8.5%) participants had both depression and anxiety. ”
Point 9. Page 8, 2nd paragraph, sentence 1 – the population was stratified according to abdominal obesity status, rather than obesity status. The term ‘adherence levels’ is incorrect since participants weren’t allocated diets to which they were expected to comply. This might be phrased as …the association of quartile membership of PDI, hPDI and uPDI with depression and anxiety was determined (Figure 2). The following sentence might be ‘Specifically, the odds ratio for depression was lower by 32% (OR 0.68, 95% CI:0.58-0.81, p<0.001) in the highest quartile of the PDI in participants without abdominal obesity compared to the lowest quartile, and…’
Reply:
Thank you for your professional comments, and we apologize for the lack of precision in our wording. Based on your suggestion, we changed “obesity status” to “abdominal obesity status” and also revised the manuscript according to your professional description. We have made great progress in this process and thank you again for your patient guidance.
Changes in the revised manuscript:
Page 9, paragraph 1, line 1: “After stratifying the population according to abdominal obesity status, we inves-tigated the association of quartile membership of PDI, hPDI and uPDI with depression and anxiety (Figure 2). Specifically, the odds ratio for depression was lower by 33% (OR = 0.68, 95% CI: 0.58-0.81, p<0.001) in the highest quartile of the PDI in participants with abdominal obesity compared to the lowest quartile, and by 49%(OR = 0.51, 95% CI: 0.41−0.64, p < 0.001) in the highest quartile of the PDI in participants without ab-dominal obesity compared to the lowest quartile. Among participants with abdominal obesity, the odds ratio for depression was lower by 36%( OR = 0.64, 95% CI: 0.54−0.75, p < 0.001) in the highest quartile of the hPDI compared to the lowest quartile and by 41% (OR = 0.59, 95% CI: 0.48−0.73, p < 0.001) in those without abdominal obesity. Among partici-pants with abdominal obesity, the odds ratio for depression was higher by 43% (OR = 1.43, 95% CI: 1.20−1.71, p < 0.001) in the highest quartile of the uPDI compared to the lowest quartile and by 80% (OR = 1.80, 95% CI: 1.44−2.25, p < 0.001) in those without abdominal obesity. The association between PDI scores and anxiety was not significant in those with abdominal obesity (p > 0.05); among participants without abdominal obesity, those in the top quartile of the PDI had a 25% (OR = 0.75, 95% CI: 0.56−0.99, p < 0.05) lower odds ratio of anxiety compared to the lowest quartile. Among participants with ab-dominal obesity, the odds ratio for anxiety was lower by 23% (OR = 0.77, 95% CI: 0.61−0.96, p < 0.05) in the highest quartile of the hPDI compared to the lowest quartile and by 48% (OR = 0.52, 95% CI: 0.39−0.70, p < 0.001) in those without abdominal obesity. Among participants with abdominal obesity, the odds ratio for anxiety was higher by 46% (OR = 1.46, 95% CI: 1.15−1.85, p < 0.01) in the highest quartile of the uPDI compared to the lowest quartile, and by 58% (OR = 1.58, 95% CI: 1.17−2.12, p < 0.01) in those without abdominal obesity. ”
Point 10. The title for Figure 2 should note that results for the fully adjusted model (i.e. Model 3) are presented. It is not clear why the results are listed vertically from Quartile 4 to Quartile 1, while the legend is listed vertically from Q1 to Q4.
Reply:
Thank you for your suggestion. We have revised the title of Figure 2. We apologize for the problem with our legend, and we have adjusted the order of the legends as soon as we realized the problem.
Changes in the revised manuscript:
Page 11:
Figure 2. ORs and 95% CIs for developing depression and anxiety by plant-based diet indices in the fully adjusted model, stratified by abdominal obesity.
Point 11. The interaction between the PDI, HPDI, uPDI and abdominal obesity for the outcomes of depression and anxiety is stated to be statistically significant (below Figure 2). Because this applies to 6 different models, the statement should be (all p<0.05, data not shown). The data in tables S2-S3 are relevant, but the statement of statistical significance for the interaction is a modelled result.
Reply:
Thank you for your professional comments. We have removed the p-values that are not fully representative of the data results here and added conclusions based on the results of the interaction analysis. However, since the interaction analysis is only a supplementary method to support our conclusions, we have only described it briefly. In addition, we also modified the description of the interaction in the discussion and conclusion sections by referring to more rigorous descriptions in other literature.
Changes in the revised manuscript:
Page 11, paragraph 1, line 1: “Overall, the interaction between the PDI, hPDI, uPDI, and abdominal obesity in their effect on the prevalence of depression and anxiety was shown to be statistically significant. The negative multiplicative effect of having abdominal obesity and the highest quartile of PDI, hPDI, and the lowest quartile of uPDI on depression and anxiety was more pronounced compared to not having abdominal obesity(all p < 0.01)(Tables S2-S3). ”
Page 12, paragraph 1, line 2: “In addition, we observed a significant interaction between the plant-based diet indices and abdominal obesity in their effect on the prevalence of depression and anxiety. ”
Page 14, paragraph 2, line 5: “There was a significant interaction between these plant-based diet indices and abdominal obesity in their effect on the prevalence of depression and anxiety. ”
Discussion:
Point 12. The authors should review the language used to avoid making inferences of causality (which the study design can’t support). For example, the first sentence of the second paragraph is a statement of causality.
Reply:
Thanks for your constructive suggestions. We have rechecked the entire text and corrected the inappropriate sentences to prevent inappropriate expressions from remaining in the text.
Changes in the revised manuscript:
Page 12, paragraph 2, line 1: “We found that a higher consumption of plant-based foods, especially healthful plant-based foods, may be able to reduce the prevalence of depression and anxiety in participants, which is largely aligned with previous epidemiological findings. ”
Point 13. The authors suggest that the difference in results by gender were small however this might be better phrased in the discussion as ‘except for the absence of an association between plant-based diet indices and anxiety in men’.
Reply:
We strongly agree with your valuable suggestion. As you said, the results of the subgroup analysis showed no significance between men and anxiety, and our statement that "there are gender differences" was not clear enough, so we have revised the sentence as you suggested.
Changes in the revised manuscript:
Page 12, paragraph 1, line 4: “Except for the absence of an association between the plant-based diet index and anxiety in men, the results of the subgroup analysis were generally consistent with the primary results. ”
Point 14. The statement about association of depression and anxiety with greater consumption of animal based food is not anchored to a reference and may be a statement from analysis done as part of the present study. However, this data is not shown and the statement isn’t particularly need and should be removed.
Reply:
Thank you for your suggestion. In fact, “greater consumption of animal-based foods is associated with higher prevalence of depression and anxiety” is our inference based on the data results in Table 2, Table 3, and Figure 2. We concluded from the data on PDI and hPDI in our study that “greater consumption of plant-based foods is associated with lower prevalence of depression and anxiety”. Also, “greater consumption of animal-based foods is associated with higher prevalence of depression and anxiety” was concluded from the uPDI data in the study. This sentence in the article is not only a summary of our data results, but also allows us to cite other relevant literature to support this idea. Therefore, we have not removed this sentence. If our understanding is inappropriate, please continue to contact us and we are willing to make changes.
Point 15. Where cited study findings are based on animals (mice), this should be stated
Reply:
Thanks for your valuable suggestions. We immediately checked the cited literature, and we clarified that the findings in the references were conclusions drawn from animal studies.
Changes in the revised manuscript:
Page 12, paragraph 2, line 25: “First, animal studies and clinical observational trials indicate that inflammation and oxidative stress may trigger the pathogenesis of depression and anxiety by affecting brain function. Plant-based foods are often rich in anti-inflammatory and anti-oxidative flavonoids and antioxidants, which can reduce inflammation and oxidative stress and can be effective in forestalling the progression of mental disorders. Second, results from mice tests and and clinical randomized controlled trials suggest that deficiencies of zinc and magnesium are not only associated with significant increases in inflammatory markers, but also have an impact on the function of the hypothalamic–pituitary–adrenal (HPA) axis, which exerts an essential effect in mental disorders such as depression. ”
Comments on the Quality of English Language
Point 16. The quality of English expression is generally very high. In a few places, word choice is not precise and meaning is not clear but this occurs in only a handful of places.
Reply:
After receiving your suggestion, we have rechecked the language of the manuscript. We hope that the revised manuscript will meet with your approval. In addition, we are full of admiration for your professional insights and rigorous scientific attitude, and we thank you again for your help to us. Wish you all the best.
